# Developing a Classification of Spinal Medical Devices: Has the Time Come? Review of the Literature and a Proposal for Spine Registries

**DOI:** 10.3390/bioengineering12080853

**Published:** 2025-08-08

**Authors:** Veronica Mari, Simona Pascucci, Andrea Piazzolla, Pedro Berjano, Michela Franzò, Letizia Sampaolo, Eugenio Carrani, Marina Torre

**Affiliations:** 1Italian National Institute of Health, 00161 Rome, Italy; letizia.sampaolo@iss.it (L.S.); eugenio.carrani@iss.it (E.C.); 2Department of Mechanical and Aerospace Engineering, Sapienza University of Rome, 00184 Rome, Italy; pascucci.1583913@studenti.uniroma1.it (S.P.); michela.franzo@uniroma1.it (M.F.); 3Azienda Ospedaliero Universitaria Consorziale “Policlinico”, 70124 Bari, Italy; dott.piazzolla@gmail.com; 4IRCCS Ospedale Galeazzi Sant’Ambrogio, 20157 Milan, Italy; pberjano@gmail.com

**Keywords:** registry, spine, taxonomy, medical device, database, implant traceability, patient safety, public health

## Abstract

Registries require standardized component libraries based on predefined taxonomies to ensure detailed and structured descriptions of implanted devices, enabling effective monitoring of implant safety. Considering the growing use of spinal implantable devices, we aimed to propose a comprehensive classification framework for spinal devices, to be integrated into the Italian Spine registry framework. The taxonomy was created using a detailed process that included reviewing existing literature, analyzing technical documents, selecting important device characteristics, obtaining feedback from manufacturers, and converting the information into a format suitable for IT systems. Our findings showed the lack of a globally accepted classification system. We identified four primary categories, further refined into subcategories, complemented by attributes for device identification, traceability, and characterization, then structured them using XSD schemas. Our proposal represents the first known attempt to implement a taxonomy for spinal implants, with the potential to serve as an international reference. A structured classification system would enhance registry interoperability, facilitate cross-registry comparability, and improve the early detection of adverse events, thereby strengthening patient safety and clinical outcomes. Furthermore, the adoption of a unified classification framework would improve surgeons’ clinical practice and support policymakers in developing early prevention strategies, ultimately improving patient care.

## 1. Introduction

The Medical Device Regulation (EU) 2017/745 (MDR) [1] establishes mandatory procedures for certifying and registering medical devices in Europe. It requires manufacturers to classify medical devices according to risk level and characteristics and meet restrictive marketing obligations. Therefore, it entails a more rigorous evaluation of clinical effectiveness and safety than in the past [2]. Furthermore, the regulation suggests encouraging the establishment of registers and databanks for specific devices, setting common principles for collecting comparable information to evaluate their safety and performance and ensuring their traceability (article 108 [1]).

Implant registries can collate extensive and structured data on implants’ performance, the surgical procedures in which they are used, and implanted patients’ quality of life. This data is used to assess devices’ effectiveness by measuring their survival in vivo and identifying patients who may require a surgery recall in case of implant failure [3,4].

Registries must be based on a comprehensive understanding of each technology’s characteristics to assess the performance of medical devices adequately. Furthermore, they must refer to an organized database with standardized technical information. Nickerson defined taxonomies as “tools for grouping objects into domains based on common characteristics” [5]; based on this definition, taxonomies represent useful tools for structuring medical device databases for use in registries as they allow for clear and unambiguous classification of a given category of devices, as well as the provision of detailed technical descriptions [6,7,8,9].

In recent decades, a considerable increase in the number of spinal surgery procedures was observed in Italy [10]. Furthermore, spinal devices are characterized by a high level of complexity and invasiveness and are classified by MDR, in numerous instances, in the highest-risk class [11]. Consequently, a notable increase in spine registries over time was observed [12]. The need for spinal registries to define common standards to collect clinical and implant data was emphasized on 23 March 2023, at the 1st International Meeting of Spinal Registries held at the Royal National Orthopaedic Hospital of Stanmore (UK) [13]. This meeting marked the launch of an international working group aimed at defining policies and standards for spinal registries.

In 2020, a new project started in Italy to establish the methodological basis for the future national Spine Registry, which includes defining the minimum data set of variables to be collected and designing a structured classification system (taxonomy) for spinal devices.

This work aims to summarize the knowledge available in the literature on spinal device taxonomies and to present a proposal for such a taxonomy tailored to the Italian context as a basis for a further extension to the international level.

## 2. Materials and Methods

### 2.1. Existing Medical Device Nomenclatures

The following medical device nomenclature systems are currently in use worldwide:-EMDN—European Medical Device Nomenclature: Adopted by the European Union for regulatory purposes, in particular to classify devices within the European database (Eudamed). It is freely accessible and has a hierarchical structure based on the Italian national medical device classification (CND), developed in Italy in 2007 by the Ministry of Health (MoH) [14,15].-GMDN—Global Medical Device Nomenclature: Provides a single, standardized nomenclature for the global classification of medical devices. It is used internationally, particularly in the Food and Drug Administration (FDA) and Medical and Healthcare Products Regulatory Agency (MHRA) regulatory environment. It was generated by exploiting existing classifications. Access is chargeable. Its structure is based on numeric codes linked to standardized terms and definitions describing the generic types of medical devices [16,17].-SNOMED CT—Systematized Nomenclature of Medicine Clinical Terms: Enables interoperability through the use of common clinical terminology. It is used internationally, mainly in the clinical health sector to represent clinical information. SNOMED CT includes device-related concepts and is partially mapped to GMDN. Access is by licence and the structure is of an ontological nature [18,19].

### 2.2. Italian Framework

The Italian MoH has invested heavily into the monitoring of the safety of medical devices over the last decades. To this end, besides CND, in 2007 the MoH created a national database of medical devices [20] structured according to CND. Since then, manufacturers have been required to register their products in this database in order to market them in Italy.

To monitor the safety of implantable devices, on 3 March 2017 the National Registry of Implantable Prostheses (RIPI) was established by a Prime Ministry’s Decree [21] at the National Institute of Health (ISS). RIPI is organized as a modular registry system, where each registry considers a specific device category with a substantial impact on public health [22]. The National Spine Registry (RIDIS) is designed as a module of RIPI [23].

### 2.3. Taxonomy Design

The following four steps were performed to select medical device (MD) categories and sub-categories, as well as the related technical characteristics (attributes) to be included in the spinal device taxonomy:Review of the literatureA comprehensive online search was conducted by LS to identify the taxonomies used by spinal registries. MEDline/PubMed, Biological Science Collection (ProQuest), Scopus, Web of Science (Science and Social Science databases, Science and Social Sciences Conferences), and grey literature databases were searched; no publication language limitation was set (the search strategies are available as Appendix A). Records were deduplicated by VM and MT, who then screened and selected the articles or reports based on their titles and abstracts. The remaining articles or reports were then assessed for eligibility by full-text reading.Articles and reports were excluded if they: (1) did not relate to spinal devices, except for those that considered taxonomies in named active national registries; (2) classified pathologies instead of devices; and (3) were not retrievable in English as full text.Technical document analysis and selection of MD attributesThe EMDN was analyzed, with the codes of categories related to spinal devices being selected as a result. The Italian national MoH MD database was queried using the selected EMDN codes to identify the registered spinal devices. Based on the information retrieved for these devices, the web was extensively browsed to collect technical datasheets, manuals, and catalogues related to them and to any other available spinal device. All these documents were thoroughly examined by SP, MF, and MT to identify the attributes associated with each category (e.g., shape, geometry, dimension, material, spinal part involved), following a maximum inclusion criterion. The clinical experts (AP and PB) discussed the identified attributes to select the ones essential for registry analyses (i.e., health outcome assessment, traceability, and MD safety and performance measurement), that were then organized into a taxonomy.Assessment by manufacturersMD specialists from the industry (NuVasive—San Diego, CA, USA, recently acquired by Globus Medical; Stryker Corporation—Kalamazoo, MI, USA; Zimmer Biomet—Warsaw, IN, USA; DePuy Synthes (Johnson & Johnson MedTech)—New Brunswick, NJ, USA; Globus Medical—Audubon, PA, USA; and Exactech—Gainesville, FL, USA) were invited to assess the taxonomy already defined with the clinicians, to integrate it with further information, and to validate it. Their task was to classify all the attributes as either “available”, “not available”, or “searchable”. This process was undertaken to identify which attributes could be uploaded to the future library with minimal effort. After discussion, only the attributes classified either as “available” or “searchable” were included and the definitive version of the taxonomy was released.IT implementationThe taxonomy was structured, modelled and encoded by VM and EC into XSD (“eXtensible markup language Schema Definition”) schemas to properly facilitate the exchange of information in an XML language for future implementation in IT platforms.

## 3. Results

The literature review was conducted following the strategies described in the Appendix A. Initially implemented in 2020 and then updated in 2024 to check for new findings, it yielded 200 articles/reports. After removing duplicates (*n* = 70), 130 articles/reports remained for title and abstract screening. Eight articles/reports were considered eligible for full-text review after screening, but one was not available in English and, therefore, was excluded. Finally, seven documents were included in the analysis. The review process is presented as a flowchart in Figure 1.

The seven selected documents (six peer-reviewed papers [6,24,25,26,27,28] and one report of the Department of Health and Human Services on the US Federal Register [29]) described and discussed the following: the process to the setting up of a taxonomy for an arthroplasty registry and the identification of the core set of attributes considered [6]; the importance of accurate device classification as a crucial tool for surveillance and the assessment of comparative effectiveness [24]; the need to reclassify these types of devices in the US context [29]; and the characterization of specific types of spinal devices such as the Growth Friendly Spine Implants [25], implants for the reconstruction of the anterior and middle spinal columns [26], or posterior dynamic stabilization devices [27,28]. However, a comprehensive taxonomy of the different types of spinal devices was not described in any of the documents identified.

Exploration of the EMDN showed that spinal surgery devices were referenced to the following codes: P0907 “Spine Stabilization Prostheses and Systems” and K0103 “Spinal Endotherapy Devices”. Thorough web searches resulted in a large number of datasheets, manuals, and catalogues that supported the definition of a hierarchical classification. The following four categories of devices were identified: (1) Fixation systems. (2) Cages. (3) Discal prostheses. (4) Augmentation systems and fillers (Table 1).

Figure 2 shows the multi-level structure of the designed taxonomy with descriptions of each identified category and its associated sub-categories. Each sub-category refers to a type of medical device that can be described by a core set of technical/functional and identification/traceability attributes. The selected technical and functional attributes (blue lines) are specific to the sub-category because they depend on its characteristics. In contrast, the identification and traceability attributes (green lines) are common to all sub-categories because they are derived from regulatory requirements.

Among the invited manufacturers, NuVasive responded and actively participated in the discussion. The following tables (Table 2: Fixation systems; Table 3: Cages; Table 4: Discal prostheses; Table 5: Augmentation systems and fillers) summarize the selected technical and functional attributes to be included in the final taxonomy for each category which are beneficial for the registry purposes and feasible to be provided by manufacturers.

**Figure 2 bioengineering-12-00853-f002:**
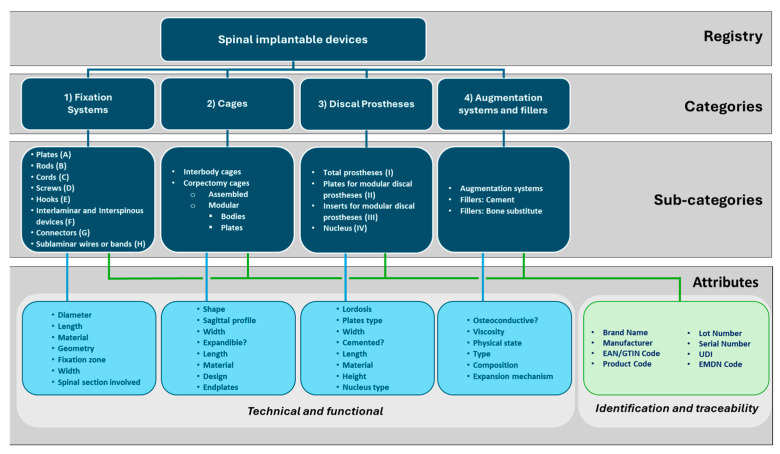
Hierarchical classification of spinal implantable devices and related attributes (in light blue: core sets of technical and functional attributes; in green: core sets of the identification and traceability attributes). Note: categories are numbered according to what is reported in Table 1; sub-categories are numbered according to what is reported in Table 2 for the Fixation Systems and in Table 4 for the Discal Prostheses.

Figure 3 shows the taxonomy represented on an XSD schema as an example applied to RIDIS for the category “Fixation Systems”. In particular, this category is exploded into its eight sub-categories and then the sub-category “Connectors” is detailed with all the identified technical and functional attributes.

## 4. Discussion

The World Health Organisation (WHO) states that “a standardized classification and nomenclature of medical devices will serve as a common language for recording and reporting medical devices across the whole health system at all levels of health care for a whole range of uses” [30].

In addition, over the last six years the WHO has assessed the existing MD classifications by comparing their peculiarities and constraints. It emerged that SNOMED CT is not primarily focused on medical devices but includes device-related concepts and is partially mapped to GMDN. In conclusion, the WHO decided not to develop a new classification and to use the EMDN and GMDN systems. Moreover, they encouraged the universal adoption of these systems to facilitate standardization and convergence in the field, and their integration into global healthcare systems [19,31].

However, these classification systems lack sufficient detailed information to fulfil the objectives of a registry, as they do not contain attributes which a taxonomy does.

This work aimed to summarize the knowledge available in the literature on the definition of taxonomies characterizing spinal devices and to present a first proposal for a structured classification of spinal devices to be implemented within the Italian Spine Registry and to be shared internationally.

Implantable prosthesis registries have proliferated worldwide in recent decades [32,33], as they have proven to be an effective method of the early detection of implant failure [4]. The importance of registries is even more recognized when considering high-risk implantable MDs, given their high impact on healthcare costs and their invasiveness for patients [34,35]. MDR was designed to improve patient safety and prevent severe incidents such as those related to Poly Implant Prothèse (PIP) breast implants and Metal-on-Metal (MoM) hip prostheses [36,37] that occurred in 2010 and affected thousands of patients worldwide. Indeed, article 108 of the MDR requires Member States to take appropriate measures to encourage the establishment of registries or databanks [1] to monitor implantable medical devices. Spinal implants belong to the highest risk classes, not only according to MDR classification rules but also according to the Australian government provisions [38] and the FDA [29].

MD registries are recognized to be a useful tool for monitoring the safety and performance of medical devices. They can provide scientific evidence for clinicians to improve their clinical practice, for manufacturers to conduct effective post-marketing surveillance, and for notified bodies to allow them to meet regulatory requirements [39]. Registries could help to overcome the unintended consequences of the MDR’s implementation. Shortages of medical devices may occur as several manufacturers intend to remove their products from the market due to the high costs of conducting clinical investigations to collect sufficient evidence to assess or confirm MDR conformity, as required by the regulation [40,41]. Recently, it was highlighted that some catheters, which are widely used worldwide for fetoscopic endoluminal tracheal occlusion, were not CE-recertified following the MDR’s entry into force. As an alternative device is unlikely to be available before 2026, fetuses and their parents are currently prevented from accessing this recognized life-saving procedure [42]. Concerns have also been raised about the markets for orthopedic implants [43] and endoscopes [41]. Hopefully, the new EU measures that oblige manufacturers to notify the relevant authorities, health institutions, healthcare professionals, and economic operators of any interruption to the supply of critical medical devices will help to avoid serious consequences for patient health [44].

In UK, the Independent Medicines and Medical Devices Safety Review recommended that “A central patient-identifiable database should be created by collecting key details of the implantation of all devices at the time of the operation. This can then be linked to specifically created registers to research and audit the outcomes both in terms of the device safety and patient reported outcomes measures.” [45]. National registries are essential tools to protect patients’ health and safety, and support recalls in the event of an accident. To assess the effectiveness and safety of implantable devices, they need to identify and characterize implanted medical devices accurately. This requires access to structured and comprehensive MD libraries. Such libraries integrated into the registry infrastructure can support healthcare professionals in data collection by reducing registration burden and identification errors, especially when the device is selected by scanning its barcode. They can also support reporting comparative analyses on device performance related to the attributes described by the associated taxonomy.

Taxonomies are recognized as valuable tools in different contexts. For example, Henschke et al. created a taxonomy of reimbursement requirements for HTA assessment [46]. Vannelli and Visintin highlighted the use of taxonomies by manufacturers to identify competitive or complementary solutions and reported a taxonomy model that links devices to diagnostic and treatment processes [47]. According to the results of our literature review, no comprehensive taxonomy for spinal implants has been defined to date, and few peer-reviewed papers dealing with this topic were published before 2020. Therefore, this study aimed to bridge this gap by developing a taxonomy proposal, considering technical and functional attributes useful for the further comparative evaluation of spinal implants. Good examples of this approach in the field of joint replacement are the implementation of the International Prostheses Library (IPL) promoted by the International Society of Arthroplasty Registries (ISAR) [48], the National Joint Registry (NJR) Component Library [49], and the Endoprothesenregister Deutschland (EPRD) Product Library [7]. Both the NJR and EPRD libraries have been recently harmonized to make the two registries interoperable.

With no previous experience to draw on, the multidisciplinary team’s work proved essential in organizing the vast number of documents found on the web. It allowed clinical and technical perspectives to be considered in defining a draft version of the taxonomy. Discussion with industry was essential to detail the technical information needed to characterize each device and to verify the feasibility of populating the framework that was being defined. Indeed, to ensure the high quality of information on registered medical devices, manufacturers should be responsible for feeding the MD libraries. This considerable effort will be rewarded by the benefits derived from the registry analyses to meet the post-market surveillance requirements set by the MDR. The industry must be actively involved in taxonomy design to test the feasibility of fully populating the database under development. Therefore, the definition of the taxonomy should be a compromise between the desire to have a very detailed description of the device and the actual availability of this information. In this phase, a maximum inclusion approach was used to select the attributes, with all attributes described as “available” or “searchable” being included in the taxonomy. However, as only one industry was involved in this study, it may be useful to discuss this proposal at a larger table on an international level.

The Orthopaedic Data Evaluation Panel (ODEP) is an independent panel of experts that provides objective ratings of the strength of evidence available on the performance of medical implants and is recognized as a reliable source of information on joint replacement implants. They have recently published the Methodology for Spine Implants [50] and launched the International Spine Registry working group (ISR) to build consensus among spine registries all around the world and other key stakeholders (regulators, clinicians, notified bodies, and MD manufacturers) on the use of common standards for valuable data collection for the further assessment of spinal devices and outcome comparison on an international level. The ISR focuses on the following three main topics: (1) the definition of the Minimum Data Set (MDS) that would be recommended for collection in spine registries internationally, (2) which Patient-Reported Outcome Measures (PROMs) should be recommended for collection following the suggestions made by the international collaboration led by International Consortium for Health Outcomes Measurement (ICHOM) and, finally, (3) the definition of an implant classification architecture that specifies the attributes to be collected for implantable devices. The basic idea is to agree on a common language between registries so that data from different sources can be aggregated and analyzed with greater statistical power and worse-than-expected outcomes can be detected earlier. An ISR-led survey of spinal registries highlighted a lack of uniformity in the implant data collected [51,52] and the need to move towards a single taxonomy shared by all registries and agreed with industry. In the future, this taxonomy could be used to implement a single implant database fed directly by manufacturers, following the example of IPL and NJR-EPRD for joint arthroplasty. This solution could lead to an optimization of the costs and resources required to maintain such a detailed and comprehensive database. Moreover, it will enable registries to compare their data internationally, increasing the number of observed events and improving the ability to earlier detect potential problems. This will prevent further patients from being harmed by these implants. In Italy, the ODEP rating has been used as a reference in some regional procurement tenders for joint prostheses, where it is used to assess the quality of the supplied devices [53].

Therefore, if registries adopt more precise product and outcome descriptions, they will be able to help manufacturers obtain the ODEP rating, enabling them to achieve better scores when participating in procurement tenders. This process has economic implications for manufacturers, who have a better chance of winning the tender, and for the public health system, which can choose higher-quality devices. This approach is increasingly being used for joint replacements and is likely to become standard in the near future for spinal surgery devices too.

Particular attention has been paid to designing this taxonomy as a modular structure that can easily integrate future updates to the selected categories, subcategories, and attributes, if required. An IT structure is crucial for a registry to fulfil its purpose and to process massive data volumes. To this end, adopting common standards based on recognized software engineering principles is essential. In order to allow the immediate implementation of this proposal, the taxonomy translation into a computer language is presented, adopting the XML language. Using the XML language, based on XSD schemas, is a consolidated approach in data exchange frameworks. The most valuable characteristics of XML are (i) the open and non-proprietary standard, (ii) the ease of reading data by automated systems, and (iii) the power of data transfer. The use of this language in MD registries is innovative. Therefore, its translation into XML format has been presented to facilitate the IT implementation of the defined taxonomy in the context of future international collaborations to support interoperability among different health information systems (e.g., Electronic Health Records) and different spine registries. Following this approach, manufacturers will be required to structure their data using the XML format, per the XSD schemas, to feed the library.

Finally, the increasing use of electronic systems for health data collection highlights the need for standardized and unambiguous systems for naming and coding medical devices [18]. The adoption of a common description of attributes and recognized standards for data collection is a necessary condition for further international cooperation.

### 4.1. Limitations

There are some limitations to this study. Although a thorough search was carried out to identify all spinal devices, some might not have been included.

A first limitation may be the use of EMDN as a starting point for subsequent research. Other classifications/nomenclatures may have revealed the existence of other types of devices not covered by EMDN. Possible gaps in EMDN could have resulted in some spinal devices not being appropriately classified in the Italian national database and therefore excluded from our query (misclassification). Another limitation could be that the Italian national database only includes devices marketed in Italy. Therefore, devices used in other countries may not have been included. In addition, some devices may have been excluded from the final list due to the lack of expertise of the technical panel responsible for the selection (selection bias). Another important limitation is the possible exclusion in the presented taxonomy of some MD attributes related to confidential information (i.e., protected by patents) and, therefore, not shareable by manufacturers. Finally, although all the manufacturers active on the Italian market were invited to participate, only one responded, thus limiting the inclusion of attributes otherwise considered by the other manufacturers.

### 4.2. Future Developments

The taxonomy presented in this paper is a first proposal that will be further improved through a broad international discussion to increase its usefulness and acceptance by the global medical community, to ensure its global relevance, and to reduce regional biases. This could be performed within the ISR, which considers this work as a useful reference to define a common infrastructure for the registries to ensure the traceability and characterization of the implanted devices. In particular, in this context we expect that the proposed taxonomy could be analyzed by a larger number of manufacturers marketing spinal devices worldwide, an approach that would ensure comprehensive coverage of device attributes. In addition, broader manufacturer engagement would mitigate potential biases from limited input and ensure a more robust and universally applicable taxonomy. Finally, it will also be possible to validate the selected attributes by also applying inter-rater reliability measures.

In Italy, the Competent Authority has invested in recent years in the establishment of MD registries and in the publication of national provisions requiring manufacturers to provide the registries with the necessary information to ensure MD traceability. We believe that, following these requirements and the increasing application of the MDR, industry would be more interested in further discussions with the registries.

Once the Italian Spine Registry is established and starts collecting data, a permanent multidisciplinary technical panel will monitor the functioning of the registry. Thus, it will periodically review and update the taxonomy according to the standards defined at the international level and through continuous discussion with industry to incorporate the latest innovations, support ongoing regulatory compliance and improve clinical practice.

Finally, the RIDIS data collection system was designed to use Hospital Discharge Records (HDRs), supplemented by an additional Minimum Data Set (MDS). In addition to information relating to the patient and the procedure, the MDS also includes data essential for MD traceability and technical MD characterization, according to the taxonomy proposed in this study. Since HDRs also collect DRG codes, once the RIDIS flow is implemented it will be possible to correlate different types of implanted device with the DRG associated with hospital admission. This will provide decision-makers with additional insight to help them assess the appropriateness of the DRG and related reimbursement, as well as its association with outcomes.

## 5. Conclusions

This study has proposed a structured and comprehensive classification of spinal devices. To our knowledge, this is the first attempt to implement a taxonomy for such devices. The Italian Spine Registry will adopt this taxonomy to implement its own MD library, which is crucial for their traceability and assessment.

Reliable and internationally standardized data from national spine registries will be essential in the coming years. The goal of the International Spine Registry is to create a common global MD database, and the results of this study will hopefully serve as a reference.

Registries are recognized as a valuable tool for the early identification of devices at high risk of failure [54]. Using a common language between registries would allow comparability and dramatically increase the number of events observed, maximizing the ability to detect adverse events in advance. It would be a new process through which surgeons could improve their clinical practice, and policymakers could implement early prevention strategies with clear benefits for the quality of care provided to patients.

We hope that, thanks to the collaboration with ISR, the taxonomy we have described will be consolidated and aligned with international policies and standards to facilitate its global adoption. Ensuring regulatory compliance, supporting international interoperability, and enhancing the credibility of the taxonomy would be key benefits of this alignment.

## Figures and Tables

**Figure 1 bioengineering-12-00853-f001:**
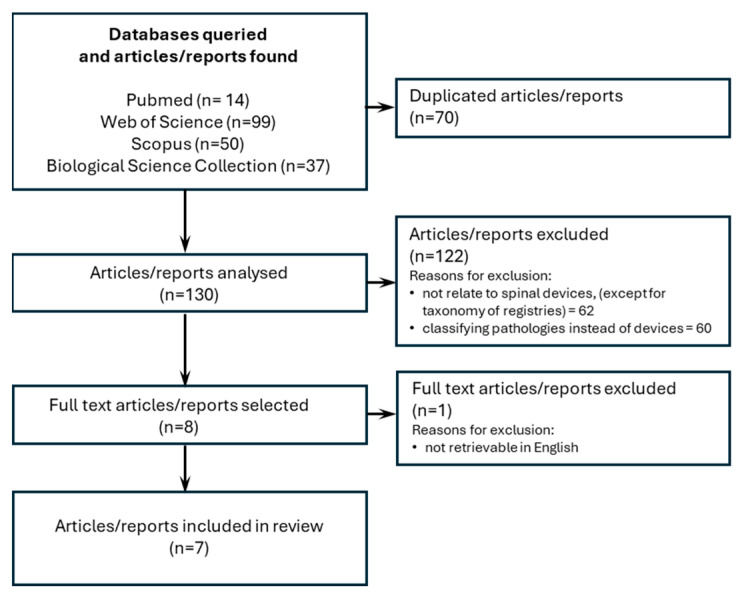
Literature review: studies inclusion flow chart.

**Figure 3 bioengineering-12-00853-f003:**
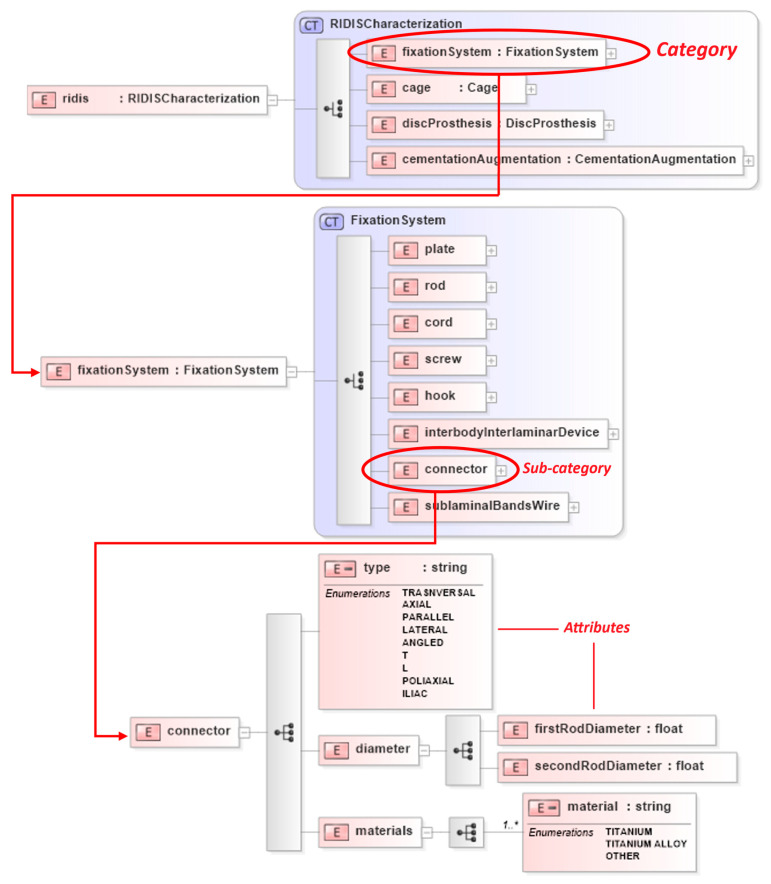
XSD schema of taxonomy applied to the Fixation Systems category.

**Table 1 bioengineering-12-00853-t001:** Selected EMDN category and type codes and descriptions for spinal implants and categories identified for registry taxonomy.

EMDN Category Code	EMDN Category Description	EMDN Type Code	EMDN Type Description	Category of Registry Taxonomy
P0907	Spine Stabilisation Prostheses and Systems	P09070301	Cervical fixation systems	(1) Fixation Systems
		P09070302	Thoracolumbosacral fixation systems	
		P09070304	Spine fixation system connectors	
		P09070399	Implantable spinal stabilisation or fixation systems—other	
		P090799	Spinal prostheses and stabilisation systems others	
		P09070101	Cages	(2) Cages
		P09070199	Spinal Fusion system—other	
		P09070201	Disc Prostheses	(3) Discal Prostheses
		P090780	Spinal stabilisation prostheses and systems—accessories	
K0103	Spinal Endotherapy Devices	K010301	Spinal percutaneous plastic devices with balloon	(4) Augmentation systems and Fillers
		K010399	Spinal endotherapy devices—other	

**Table 2 bioengineering-12-00853-t002:** Fixation Systems (1): technical and functional attributes by sub-category.

Attribute	Sub-Category (*)							
	A	B	C	D	E	F	G	H
Spinal level	✓	✓	-	✓	✓	✓	-	-
Vertebra/bone fixation area	✓	-	-	✓	-	✓	-	✓
Material	✓	✓	✓	✓	✓	✓	✓	✓
Coating material	-	-	-	-	-	✓	-	-
Mechanical characteristics	-	✓	-	-	-	✓	-	-
Type	✓	✓	-	✓	✓	-	✓	✓
Length (mm)	✓	✓	-	✓	-	-	-	✓
Thickness (mm)	✓	-	-	-	-	-	-	✓
Width (mm)	-	-	-	-	-	-	-	✓
Diameter(s) (mm)	-	✓	-	✓	-	-	✓	✓
Shape	✓	✓	-	-	✓	-	-	✓
N. of spinal levels	✓	-	-	-	-	-	-	-
Hole number	✓	-	-	-	-	-	-	-
N. of wires/bands	-	-	-	-	-	-	-	✓
Screw head type	-	-	-	✓	-	-	-	-
Screw stem type	-	-	-	✓	-	-	-	-
Screw mobility angle (°)	-	-	-	✓	-	-	-	-
Hook offset	-	-	-	-	✓	-	-	-
Hook blade	-	-	-	-	✓	-	-	-
Hook groove	-	-	-	-	✓	-	-	-
Size	-	-	-	-	✓	✓	-	-
Tightening system	✓	-	-	✓	-	✓	-	-
Compatibility with devices of other companies	-	-	-	-	-	-	-	✓

Note: ✓ = attribute available for the sub-category. (*) Legend: (A) plates, (B) rods, (C) cords, (D) screws, (E) hooks, (F) interlaminar and interspinous devices, (G) connectors, and (H) sublaminar wires or bands (including related connectors).

**Table 3 bioengineering-12-00853-t003:** Cages (2): technical and functional attributes by sub-category.

Attribute	Sub-Category			
	Interbody Cages	Corpectomy Cages		
		Assembled	Modular	
			Bodies	Plates
Spinal level	✓	✓	✓	✓
Material	✓	✓	✓	✓
Osteointegration material	✓	✓	✓	✓
Surface coating	✓	✓	✓	✓
Dimensions (L × W, Φ)	✓	✓	✓	✓
Height anterior/posterior (min/max)	✓	✓	✓	✓
Expandible (height/lordosis)	✓	✓	✓	-
Shape	✓	✓	✓	✓
Sagittal profile	✓	✓	-	✓
Surface characteristics	✓	✓	-	✓
Endplates curvatures	✓	✓	-	✓
Lordosis/kyphosis angle (°)	✓	✓	-	✓
Fixation system	✓	✓	✓	✓

Note: ✓ = attribute available for the sub-category.

**Table 4 bioengineering-12-00853-t004:** Discal prostheses (3): technical and functional attributes by sub-category.

Attribute	Sub-Category (*)			
	I	II	III	IV
Spinal level	✓	✓	✓	✓
Material	✓	✓	✓	✓
Coating material	✓	✓	-	✓
Type	-	✓	-	✓
Width (mm)	✓	✓	✓	✓
Length (mm)	-	✓	✓	✓
Height (mm)	✓	✓	✓	✓
Lordosis angle (°)	✓	✓	-	✓
Flexion/Extension (mm)	✓	✓	-	-
Lateral bending (mm)	✓	✓	-	-
Axial rotation (°)	✓	✓	-	-
To be cemented	✓	✓	-	-
Possibility to use screws	✓	✓	-	-

Note: ✓ = attribute available for the sub-category. (*) Legend: (I) total prostheses, (II) plates for modular discal prostheses, (III) insert for modular discal prostheses, and (IV) nucleus.

**Table 5 bioengineering-12-00853-t005:** Augmentation systems and fillers (4): technical and functional attributes by sub-category.

Attribute	Sub-Category		
	Augmentation Systems	Fillers: Cement	Fillers: Bone Substitute
Material	✓	-	-
Type	✓	-	-
Length release (mm)	✓	-	-
Length close stent (mm)	✓	-	-
Maximum expansion length (mm)	✓	-	-
Maximum volume (mL)	✓	-	-
Maximum pressure (bar/atm)	✓	-	-
Expansion mechanism	✓	-	-
Physical state	-	✓	✓
% Polymethylmethacrylate	-	✓	-
% Zirconium	-	✓	-
% other components	-	✓	✓
Antibiotics	-	✓	✓
Internal structure (scaffold, trabecular, other)	-	-	✓
Osteoinductive/osteoconductive	-	✓	✓
Viscosity	-	✓	-
Polymerization time (s)	-	✓	-
Polymerization mode	-	✓	-

Note: ✓ = attribute available for the sub-category.

## Data Availability

The original contributions presented in this study are included in the article/Appendix A. Further inquiries can be directed to the corresponding authors.

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
