# Peer review of "Developing a Classification of Spinal Medical Devices: Has the Time Come? Review of the Literature and a Proposal for Spine Registries"

_bioengineering, 2025, doi:10.3390/bioengineering12080853_

Round 1

Reviewer 1 Report (Previous Reviewer 2)

Comments and Suggestions for Authors

The changes do make the article a lot easier to follow and the scientific clarity is also much improved.

Author Response

Comment 1: The changes (figures and tables can be improved) do make the article a lot easier to follow and the scientific clarity is also much improved.

Response 1: Thank you for your suggestion, which has helped us to improve the presentation of information in the figures and tables. We have updated these and hope that they now convey the message to readers more effectively. We have also taken this opportunity to update the description of the EMDN codes to the latest published version, and to ensure consistency in the description of the selected categories throughout the entire manuscript.

Reviewer 2 Report (New Reviewer)

Comments and Suggestions for Authors

Authors present a review article on development of novel classification of spinal medical devices. This manuscript lacks criticism on EU Medical Device Regulation, which has made substantial problems for surgeons who use various devices, for example endoscopes. Overregulation in EU has led to increased cost of registration and extension of licence, therefore many products are after some time not available. Furthermore, manuscript has a subject which is too broad and it is not clear what is their intention - what does this new classification bring? Maybe take a clinical example and apply the classification in correlation to DRG - does this classification has any economic relevance? This seems to be a revision of the manuscript as some parts are highlighted. 

Author Response

Comment 1: Authors present a review article on development of novel classification of spinal medical devices. This manuscript lacks criticism on EU Medical Device Regulation, which has made substantial problems for surgeons who use various devices, for example endoscopes. Overregulation in EU has led to increased cost of registration and extension of licence, therefore many products are after some time not available. 

Response 1: Thank you for your comment that raises the important issue of the possible MD shortage due to the strict requirements stated by MDR. The difficulty that manufacturers face in obtaining the necessary data to support device certification is indeed one of the reasons for these shortages. Registries can produce this data by collecting information from across the country, thereby supporting manufacturers and notified bodies in meeting the MDR's requirements. Defining a device taxonomy is essential in order to enable a registry to carry out appropriate analyses, compare implanted devices, and demonstrate their safety and efficacy. We have included a paragraph referring to this issue in the discussion and have integrated the relevant bibliographic references. (See lines 255-273).

Comment 2: Furthermore, manuscript has a subject which is too broad and it is not clear what is their intention - what does this new classification bring?

Response 2: Thank you for your question. Indeed, the classification system supports the analyses carried out by the registry. It is hoped that, in the future, the spinal device registries set up in various countries will be able to refer to a single, shared classification. This would enable them to compare their data internationally, increasing the number of observed events and therefore improving the ability to detect possible implant problems earlier, thus preventing them from being implanted in further patients. We have added in the discussion a sentence referring to this perspective. (See lines 336-338).

Comment 3: Maybe take a clinical example and apply the classification in correlation to DRG - does this classification has any economic relevance?

Response 3: Thank you for your helpful comments, which have enabled us to improve our manuscript. We have added some sentences to the Discussion referring to how the classification may be correlated to DRG and its potential impact on public health expenditure. As the classification is primarily intended for manufacturers who will be required to upload all data to the MD database of the registry in the future, we preferred to limit the presentation to the technical aspects and not focus on clinical cases. (See lines 407-415 and 339-347).

Comment 4: This seems to be a revision of the manuscript as some parts are highlighted. 

Response 4: Thank you for your comment, you are correct. The manuscript that you revised was indeed a resubmission, and we were asked to track the changes between this version and the previous one.

Reviewer 3 Report (New Reviewer)

Comments and Suggestions for Authors

This paper presents a new classification of medical devices for spinal surgery. 

I have some point about the literature search: I don't see the key words used for the search. Moreover, the PRISMA guidelines are for systematic reviews including RCTs. Did the authors include only RCTs?

I think the delphi process should be better described. Was there a 100% agreement on all the items?

Author Response

Comment 1: This paper presents a new classification of medical devices for spinal surgery. I have some point about the literature search: I don't see the key words used for the search.

Response 1: Thank you for your comment. A detailed description of the search strategy was presented in the supplementary material (see line 113). Following your comment, we have also added a reference to the supplementary material in the Results section, in order to better address the reader. (See lines 147-148).

Comment 2: Moreover, the PRISMA guidelines are for systematic reviews including RCTs. Did the authors include only RCTs?

Response 2: We thank the reviewer for their useful comment, which helped us to avoid any potential misunderstandings on the part of the reader. Although we did not carry out a systematic literature review, we felt it was important to demonstrate how the review was conducted and to provide a graphical representation of the different phases. We have revised the diagram by removing all references to PRISMA and updated the manuscript text accordingly. (See figure 1, lines 153 and 157 of the text and the List of abbreviations).

Comment 3: I think the delphi process should be better described. Was there a 100% agreement on all the items?

Response 3: We thank the reviewer for the comment that enabled us to refine the method description (see line 140). Additionally, we have added a sentence to the Discussion section to highlight that the approach taken in selecting attributes was one of maximum inclusion, ensuring that all attributes described as "available" or "searchable" were included in the taxonomy. (See line 309-311).

Round 2

Reviewer 2 Report (New Reviewer)

Comments and Suggestions for Authors

Sufficient response

This manuscript is a resubmission of an earlier submission. The following is a list of the peer review reports and author responses from that submission.

Round 1

Reviewer 1 Report

Comments and Suggestions for Authors

I don't recommend publication in an international journal, due to the variation in nomenclature in different regions of the world. So, in my opinion, it should be published locally by the author to help with health policies in relation to procedures and implants.

Author Response

Comment 1: I don't recommend publication in an international journal, due to the variation in nomenclature in different regions of the world. So, in my opinion, it should be published locally by the author to help with health policies in relation to procedures and implants.

Response 1: Thank you for pointing this out. We understand your comment that helped us to improve the text. Indeed, our work was started to create a reference for the Italian Spine Registry. However, to set up the taxonomy for our registry we referred to the international context. First, we carried out a scoping review to highlight the existing registries worldwide that was published in 2023 (Pascucci et al) where it emerged that none of the existing registries had developed or published a device taxonomy. Furthermore, we enlarged and focused the literature review on the specific topic of the taxonomy. The results of this review showed that, up to date, no comprehensive taxonomies for spinal devices exist. As reported in our manuscript, to design the taxonomy we considered as a first attempt the European Medical Device Nomenclature (EMDN) currently adopted at the European level within the EUDAMED database by all the Member States. Finally, our study on the taxonomy was presented at the kick off meeting of the International Spine Registry (ISR) working group held on the 23 March 2023 at the Royal National Orthopaedic Hospital of Stanmore (UK). ISR is an initiative that has been promoted worldwide by the Orthopaedic Data Evaluation Panel (ODEP) aiming at defining policies and standards for spinal registries. Our presentation was considered as a useful reference to define a common infrastructure for the registries to ensure traceability and characterization of the implanted devices.

Following your comment, we have integrated the paper with some additional information. In particular:

  • When we present the results of our literature review, we specified that none of the selected articles include a comprehensive spinal device taxonomy (see lines 167-169).
  • In the Discussion, we have integrated the content on ODEP and added a paragraph describing the aims of the International Spine Registry initiative (see lines 290 - 309).

Reviewer 2 Report

Comments and Suggestions for Authors

The proposed classification framework is a significant step towards improving the organisation and traceability of spinal devices. However, broader international collaboration, manufacturer engagement, continuous updates, clear language, and policy alignment would likely enhance its utility and acceptance in the global medical community.
The authors discuss the proposed taxonomy's relationship with existing frameworks, highlighting its potential to serve as an international reference. They emphasise the importance of regulatory compliance and interoperability, indicating that the taxonomy is intended to complement and integrate with existing systems rather than replace them. This approach ensures that the taxonomy can be adopted globally, facilitating better organisation and traceability of spinal devices across different regions and regulatory environments.

General Feedback
In addition to the specific recommendations provided after, general improvements can be made to enhance the overall quality and impact of the article. Ensuring that all sections of the article are written in clear, straightforward language will make the content accessible to a wider audience. Considering the perspectives of various stakeholders, including clinicians, manufacturers, and regulatory bodies, will create a more comprehensive and balanced framework. Designing the taxonomy to be adaptable to future technological advancements and changes in clinical practice will ensure its long-term relevance. These general improvements will collectively contribute to the effectiveness and acceptance of the proposed classification framework in the global medical community.

Overall, the article presents a comprehensive proposal for a classification framework for spinal devices to be integrated into the Italian Spine Registry. This review aims to critically evaluate the usefulness of the proposed taxonomy and provide recommendations for improvement, considering both scientific and policy aspects, as well as clarity of language.

The proposed taxonomy aligns well with existing frameworks such as the European Medical Device Regulation (EU) 2017/745, the German Arthroplasty Registry (EPRD), and the American Joint Replacement Registry (AJRR). These frameworks focus on classification rules based on risk level and intended purpose, detailed device descriptions, and performance metrics. The proposed taxonomy shares similar approaches to technical attributes and traceability, ensuring compatibility and interoperability.

Specific Feedback
To enhance the proposed taxonomy's utility and acceptance in the global medical community, forming an international working group with experts from various countries would ensure the taxonomy's global relevance and reduce regional biases. This collaborative approach would also ensure comprehensive coverage of device attributes. Engaging more manufacturers in the development process through workshops and surveys would provide a more comprehensive understanding of the technical attributes and feasibility of the taxonomy. This broader manufacturer engagement would mitigate potential biases from limited input and ensure a more robust and universally applicable taxonomy. Establishing a periodic review process to update the taxonomy based on new findings and technological advancements would keep it current and improve clinical practice. Regular updates would incorporate the latest innovations and support ongoing regulatory compliance. Revising the article for clarity and readability, ensuring technical terms are well-defined and jargon is minimised, would improve understanding among a broader audience, including non-specialists. This approach would enhance the article's impact and accessibility. Finally, aligning the taxonomy with international policies and standards would facilitate global adoption. Ensuring regulatory compliance, supporting international interoperability, and enhancing the credibility of the taxonomy would be key benefits of this alignment.

Comments on the Quality of English Language

The article is well-written, but there are areas where clarity and readability can be improved. Here are a few specific examples with suggested revisions, but please go over the rest of the text too to address similar issues of rather complicated phrasing. 
Example 1:
  Original Text: 
"The taxonomy was developed through a systematic process comprising a literature review, analysis of technical documentation, selection of key device characteristics, assessment by manufacturers, and translation into an IT-compatible format for future implementation."
  Revised Text: 
"The taxonomy was created using a detailed process that included reviewing existing literature, analysing technical documents, selecting important device features, obtaining feedback from manufacturers, and converting the information into a format suitable for IT systems."
Comment: 
The revised text breaks down the process into simpler steps, making it easier to follow and understand.

     Example 2:
  Original Text: 
"The framework aligns with the European Medical Device Regulation (EU) 2017/745, ensuring compliance with stringent safety and effectiveness standards."
  Revised Text: 
"The framework follows the European Medical Device Regulation (EU) 2017/745, which guarantees high standards for safety and effectiveness."
Comment: 
The revised text uses more straightforward language, making the sentence clearer and more direct.

     Example 3:
  Original Text: 
"Regular updates to the taxonomy based on new findings and technological advancements in spinal devices would maintain its relevance and accuracy."
  Revised Text: 
"Updating the taxonomy regularly with new research and technological improvements will keep it accurate and relevant."
Comment: 
The revised text is more concise and easier to read.

     Example 4:
  Original Text: 
"Engage manufacturers like Nuvasive, Stryker, Zimmer Biomet, Johnson & Johnson, Globus Medical, and Exactech to classify attributes as 'available,' 'not available,' or 'searchable.'"
  Revised Text: 
"Work with manufacturers such as Nuvasive, Stryker, Zimmer Biomet, Johnson & Johnson, Globus Medical, and Exactech to categorise device features as 'available,' 'not available,' or 'searchable.'"
Comment: 
The revised text uses simpler language and corrects the spelling to UK English ("categorise" instead of "classify").

     Example 5:
  Original Text: 
"The Orthopaedic Data Evaluation Panel (ODEP) has successfully launched the 'International Spine Registry' working group to build consensus among spine registries on common standards."
  Revised Text: 
"The Orthopaedic Data Evaluation Panel (ODEP) has started the 'International Spine Registry' working group to agree on common standards for spine registries."
Comment: 
The revised text is more direct and avoids unnecessary complexity.

     Example 6:
  Original Text: 
"The selection of device attributes might be influenced by the expertise of the involved technical panel, potentially excluding some relevant features."
  Revised Text: 
"The choice of device features may be affected by the knowledge of the technical panel, possibly leaving out important aspects."
Comment: 
The revised text is clearer and more concise.

     Example 7:
  Original Text: 
"The framework aims to serve as an international reference, but the involvement of a broader, global community could enhance its applicability and acceptance."
  Revised Text: 
"While the framework is intended to be a global reference, including experts from around the world would make it more useful and widely accepted."
Comment: 
The revised text is more straightforward and easier to understand.

Author Response

The proposed classification framework is a significant step towards improving the organisation and traceability of spinal devices. However, broader international collaboration, manufacturer engagement, continuous updates, clear language, and policy alignment would likely enhance its utility and acceptance in the global medical community.
The authors discuss the proposed taxonomy's relationship with existing frameworks, highlighting its potential to serve as an international reference. They emphasise the importance of regulatory compliance and interoperability, indicating that the taxonomy is intended to complement and integrate with existing systems rather than replace them. This approach ensures that the taxonomy can be adopted globally, facilitating better organisation and traceability of spinal devices across different regions and regulatory environments.

General Feedback
Comment 1: In addition to the specific recommendations provided after, general improvements can be made to enhance the overall quality and impact of the article. Ensuring that all sections of the article are written in clear, straightforward language will make the content accessible to a wider audience. Considering the perspectives of various stakeholders, including clinicians, manufacturers, and regulatory bodies, will create a more comprehensive and balanced framework. Designing the taxonomy to be adaptable to future technological advancements and changes in clinical practice will ensure its long-term relevance. These general improvements will collectively contribute to the effectiveness and acceptance of the proposed classification framework in the global medical community.     
Overall, the article presents a comprehensive proposal for a classification framework for spinal devices to be integrated into the Italian Spine Registry. This review aims to critically evaluate the usefulness of the proposed taxonomy and provide recommendations for improvement, considering both scientific and policy aspects, as well as clarity of language.

The proposed taxonomy aligns well with existing frameworks such as the European Medical Device Regulation (EU) 2017/745, the German Arthroplasty Registry (EPRD), and the American Joint Replacement Registry (AJRR). These frameworks focus on classification rules based on risk level and intended purpose, detailed device descriptions, and performance metrics. The proposed taxonomy shares similar approaches to technical attributes and traceability, ensuring compatibility and interoperability.

Response 1: Thank you for these comments and useful suggestions. We modified the article to consider a wider audience and the different perspective of the stakeholders (as an example, see lines 245-248). The suggestions you made in this general feedback have been introduced in the revised manuscript following your specific comments.

Specific Feedback
Comment 2: To enhance the proposed taxonomy's utility and acceptance in the global medical community, forming an international working group with experts from various countries would ensure the taxonomy's global relevance and reduce regional biases. This collaborative approach would also ensure comprehensive coverage of device attributes.

Response 2: Agree. We have, accordingly, revised the text to emphasize this point and to highlight that this study represents a first attempt to design a taxonomy for spinal devices to be further improved considering both thorough discussion at the international level and contributions from further manufacturers. Our study on taxonomy was presented at the kickoff meeting of the International Spine Registry working group (ISR), an initiative promoted worldwide by the Orthopaedic Data Evaluation Panel (ODEP) aiming at defining policies and standards for spinal registries. We are now attending ISR as Italian representative, in particular contributing to the working group “Implants” that considers our proposal for the taxonomy a useful reference to define a common infrastructure for spinal registries to ensure traceability and characterization of the implanted devices.

In the Discussion, we have integrated the paragraph on ODEP, added a paragraph on ISR and a new paragraph “Future Developments”. Please see lines 290-309 and 350-361.

Comment 3: Engaging more manufacturers in the development process through workshops and surveys would provide a more comprehensive understanding of the technical attributes and feasibility of the taxonomy. This broader manufacturer engagement would mitigate potential biases from limited input and ensure a more robust and universally applicable taxonomy.

Response 3: Thank you for this comment. We agree with you, indeed all manufacturers marketing their spinal devices in Italy were initially invited to join the assessment of the taxonomy. Unfortunately, after several reminders, only Nuvasive actively contributed to the discussion. We acknowledge that this represents a limitation, but we are confident that this hindrance will be overcome in the international discussion carried out within the ISR, that sees also the participation of Industry. Referring to the Italian context, an increasing attention from the Competent Authority to the establishment of medical device registries has been observed in the last years. According to recent national provisions, to ensure medical device traceability manufacturers are required to provide the registries with the necessary information to accomplish this duty. We think that, following these requirements, Industry might be more interested in being involved in further discussions at national and international levels. We integrated these comments into the article, please see lines 344-348, 355-359 and 362-366.

Comment 4: Establishing a periodic review process to update the taxonomy based on new findings and technological advancements would keep it current and improve clinical practice. Regular updates would incorporate the latest innovations and support ongoing regulatory compliance.

Response 4: Thank you for this comment, it is a very good observation. As stated above, this is a first proposal of taxonomy to be included in the future Italian spine registry. As soon as this registry is established and starts to collect data, a permanent technical multidisciplinary panel will monitor the functioning of the registry, including periodic revisions and updating of the taxonomy, according to the standards defined at the international level and the continuous discussion with Industry.

We included these comments into the article, please see lines 366-371.

Particular attention was paid to design taxonomy as a modular structure that might easily integrate, if needed, future updates of the selected categories, sub-categories and attributes. The choice of XML language for the IT translation of the taxonomy reflects also this necessity.

We included these comments into the article, please see lines 310 - 312.

Comment 5: Revising the article for clarity and readability, ensuring technical terms are well-defined and jargon is minimised, would improve understanding among a broader audience, including non-specialists. This approach would enhance the article's impact and accessibility.

Response 5: Thank you for this comment. We revised the text paying attention to uniform technical terms and to minimize jargon.

Comment 6: Finally, aligning the taxonomy with international policies and standards would facilitate global adoption. Ensuring regulatory compliance, supporting international interoperability, and enhancing the credibility of the taxonomy would be key benefits of this alignment.

Response 6: Thank you for pointing this out. We agree with your comment, we added this concept in our conclusions (lines 387-390).

Comment 7: Comments on the Quality of English Language.

The article is well-written, but there are areas where clarity and readability can be improved. Here are a few specific examples with suggested revisions, but please go over the rest of the text too to address similar issues of rather complicated phrasing.

Response 7: Thank you for your useful input. We revised carefully the text in order to improve the readability of the manuscript and to avoid complicated phrasing, taking your examples as a reference. Referring to your examples, the n.1 has been currently integrated in the revised text (see lines 21-24). We are sorry but we couldn’t find in our manuscript the text you reported as “original” for examples 2 to 7.

Example 1:
  Original Text: 
"The taxonomy was developed through a systematic process comprising a literature review, analysis of technical documentation, selection of key device characteristics, assessment by manufacturers, and translation into an IT-compatible format for future implementation."
  Revised Text: 
"The taxonomy was created using a detailed process that included reviewing existing literature, analysing technical documents, selecting important device features, obtaining feedback from manufacturers, and converting the information into a format suitable for IT systems."
Comment: 
The revised text breaks down the process into simpler steps, making it easier to follow and understand.

Response example 1: OK

     Example 2:
  Original Text: 
"The framework aligns with the European Medical Device Regulation (EU) 2017/745, ensuring compliance with stringent safety and effectiveness standards."
  Revised Text: 
"The framework follows the European Medical Device Regulation (EU) 2017/745, which guarantees high standards for safety and effectiveness."
Comment: 
The revised text uses more straightforward language, making the sentence clearer and more direct.

Response example 2: NA

     Example 3:
  Original Text: 
"Regular updates to the taxonomy based on new findings and technological advancements in spinal devices would maintain its relevance and accuracy."
  Revised Text: 
"Updating the taxonomy regularly with new research and technological improvements will keep it accurate and relevant."
Comment: 
The revised text is more concise and easier to read.

Response example 3: NA

     Example 4:
  Original Text: 
"Engage manufacturers like Nuvasive, Stryker, Zimmer Biomet, Johnson & Johnson, Globus Medical, and Exactech to classify attributes as 'available,' 'not available,' or 'searchable.'"
  Revised Text: 
"Work with manufacturers such as Nuvasive, Stryker, Zimmer Biomet, Johnson & Johnson, Globus Medical, and Exactech to categorise device features as 'available,' 'not available,' or 'searchable.'"
Comment: 
The revised text uses simpler language and corrects the spelling to UK English ("categorise" instead of "classify").

Response example 4: NA

     Example 5:
  Original Text: 
"The Orthopaedic Data Evaluation Panel (ODEP) has successfully launched the 'International Spine Registry' working group to build consensus among spine registries on common standards."
  Revised Text: 
"The Orthopaedic Data Evaluation Panel (ODEP) has started the 'International Spine Registry' working group to agree on common standards for spine registries."
Comment: 
The revised text is more direct and avoids unnecessary complexity.

Response example 5: NA

     Example 6:
  Original Text: 
"The selection of device attributes might be influenced by the expertise of the involved technical panel, potentially excluding some relevant features."
  Revised Text: 
"The choice of device features may be affected by the knowledge of the technical panel, possibly leaving out important aspects."
Comment: 
The revised text is clearer and more concise.

Response example 6: NA

     Example 7:
  Original Text: 
"The framework aims to serve as an international reference, but the involvement of a broader, global community could enhance its applicability and acceptance."
  Revised Text: 
"While the framework is intended to be a global reference, including experts from around the world would make it more useful and widely accepted."
Comment: 
The revised text is more straightforward and easier to understand.

Response example 7: NA

Reviewer 3 Report

Comments and Suggestions for Authors

This article presents a timely and policy-relevant proposal to develop a taxonomy for spinal medical devices, aimed at improving registry-based surveillance and facilitating interoperability among national and international registries. Here are my comments;

  1. The article states that only one manufacturer (Nuvasive) provided feedback. This is a critical limitation since one viewpoint cannot generalize the feasibility of data collection across all spinal device types and companies.
  2. Please clarify, How technical features were selected and validated?  Criteria for including/excluding taxonomy attributes? Inter-rater reliability or expert agreement during classification?
  3. no mention of how this taxonomy might interface with existing electronic health records (EHRs) or international data standards.
  4. The article, largely restates background points rather than analyzing the challenges of implementation
  5. EMDN system is logical for an EU-based registry, its limitations (e.g., lack of global compatibility with FDA, SNOMED, or GMDN) are not discussed.

Author Response

This article presents a timely and policy-relevant proposal to develop a taxonomy for spinal medical devices, aimed at improving registry-based surveillance and facilitating interoperability among national and international registries. Here are my comments;

Comment 1: The article states that only one manufacturer (Nuvasive) provided feedback. This is a critical limitation since one viewpoint cannot generalize the feasibility of data collection across all spinal device types and companies.

Response 1: Thank you for this comment. We agree with you, indeed all manufacturers marketing their spinal devices in Italy were initially invited to participate in the assessment of the taxonomy. Unfortunately, after several reminders, only Nuvasive actively contributed to the discussion. We acknowledge that this represents a critical limitation. However, we highlighted in the manuscript that this paper aims to present a first proposal of taxonomy, to be used for a further wider international discussion. We are confident that the international framework might overcome this hindrance, as described in the new paragraph “Future developments”, where we have referred to the important activities that the International Spine Registry working group (ISR) is conducting, with the participation of Industry. We highlighted these aspects into the article, please see lines 344-348, 350-359.

Comment 2: Please clarify, How technical features were selected and validated?  Criteria for including/excluding taxonomy attributes? Inter-rater reliability or expert agreement during classification?

Response 2: Thank you for pointing this out and offering us the opportunity to better describe the adopted method. Indeed, we have now specified that, starting from the data sheets and brochures, attributes were selected following a maximum inclusion criterion. Moveover, we specified also the aspects essential for a registry and that industry was requested to integrate and validate the final list.

Attributes were included or excluded from the taxonomy on the basis on the concept of availability / unavailability declared by the Industry. Please see lines 123–144.

The participation of only one manufacturer represents a limit of this process. We expect that the involvement of further manufacturers within ISR will allow to consider measures of inter-rater reliability and will improve our proposed taxonomy in the future. We integrated these comments into the Discussion: see “Limitations” (lines 344-348) and “Future developments” (lines 350-361).

Comment 3: no mention of how this taxonomy might interface with existing electronic health records (EHRs) or international data standards.

Response 3: The taxonomy considers some attributes to identify each implanted device (identification and traceability attributes). This information can be usefully integrated into EHRs to associate the implanted devices with the surgery performed. We have adopted an IT language that easily allows the interoperability between spine registry and other health information systems (e.g. EHRs). Following your comment, we have integrated the text. Please see lines 322-323.

Comment 4: The article, largely restates background points rather than analyzing the challenges of implementation

Response 4: Following the useful comments received from the reviewers, we integrated the manuscript with new paragraphs that analyze also possible challenges of the taxonomy implementation. Please, see lines 295-309, 349-371, 387-390.

Comment 5: EMDN system is logical for an EU-based registry, its limitations (e.g., lack of global compatibility with FDA, SNOMED, or GMDN) are not discussed.

Response 5: Thank you for pointing this out and offering us the opportunity to describe the state of the art on existing classification systems for medical devices. We have added a paragraph in the Method summarizing their characteristics and differences (see lines 77-92), and two paragraphs in the Discussion. The first one summarizes the work done at the international level by the WHO to evaluate these systems and decide which one to adopt for its own purpose. Based on the results obtained, the WHO decided to adopt both EMDN and GMDN, to promote their wider use and to achieve standardization and convergence. The second one in the “Limitations” section. Please see lines 220-228 and 334-340. 

In our study, we used EMDN as a starting point just to identify an initial grouping of device types. Our choice is due to the opportunity to easily query the Italian national MD database, which is based on it. We integrated this concept in the Method, please see lines 94-98 and 125-126.

Round 2

Reviewer 2 Report

Comments and Suggestions for Authors

The abbreviations list does not include all abbreviations used, e.g., GMDN, to name one such.

Otherwise seems suitable to accept now

Author Response

Comment 1: The abbreviations list does not include all abbreviations used, e.g., GMDN, to name one such. Otherwise seems suitable to accept now.

Response 1: Thank you for pointing this out. We have carefully checked and revised the entire manuscript in line with your suggestions. The revisions from this second round are highlighted in yellow. In particular:

  • We have added in the text the expansion of the acronyms, if not previously reported
  • We have added the missing acronyms and their expansions cited in the text (FDA, MHRA, GMDN, ICHOM, PROMs and SNOMED CT) to the Abbreviations Table
  • For acronyms that are well known to technical readers but not to a wider audience, we have only left the acronym in the manuscript, reporting it with its expansion in the Table (IT, ISO, EAN, GTIN, UDI and HTA).
  • we did not include the following well-known acronyms in the Abbreviations Table: EU, US and UK

Reviewer 3 Report

Comments and Suggestions for Authors

The manuscript is suitable for publication in its current form.

Author Response

Comment 1: The manuscript is suitable for publication in its current form.

Response 1: Thank you for helping us to improve our manuscript.